

# Record linkage of banks and municipalities through multiple criteria and neural networks

Antonio Maratea, Angelo Ciaramella and Giuseppe Pio Cianci

Department of Science and Technology, University of Naples "Parthenope", Naples, Italy

## ABSTRACT

Record linkage aims to identify records from multiple data sources that refer to the same entity of the real world. It is a well known data quality process studied since the second half of the last century, with an established pipeline and a rich literature of case studies mainly covering census, administrative or health domains. In this paper, a method to recognize matching records from real municipalities and banks through multiple similarity criteria and a Neural Network classifier is proposed: starting from a labeled subset of the available data, first several similarity measures are combined and weighted to build a feature vector, then a Multi-Layer Perceptron (MLP) network is trained and tested to find matching pairs. For validation, seven real datasets have been used (three from banks and four from municipalities), purposely chosen in the same geographical area to increase the probability of matches. The training only involved two municipalities, while testing involved all sources (municipalities vs. municipalities, banks vs banks and and municipalities vs. banks). The proposed method scored remarkable results in terms of both precision and recall, clearly outperforming threshold-based competitors.

## INTRODUCTION

Record Linkage (RL from now on, also called *entity resolution* or *entity matching*), is the process of identifying records coming from different sources that refer to the same real-world entity. Similarly, *Record Deduplication* (RD from now on) is the process of identifying duplicate records, where the same entity of the real word has been entered multiple times in the same database. The main difference between RL and RD is that records coming from different sources may lack a common identifier and schema (please refer to *Christen (2012)* and *Zhang (2011)* for a general discussion of related issues).

The pairing of records or the identification of duplicates is a statistically challenging and computationally demanding problem: scarce data quality control results in errors, noise, missing values, omissions, ambiguities and even lies in the data, that combined with the differences in database schemas and regional conventions when the number of nationalities grows, results in a daunting number of factors to be considered. The brute-force comparison All Versus All (AVA) of the records from different sources to discover occasional matches is unfeasible even for a modest volume of data. Notwithstanding these

Corresponding author
Antonio Maratea,
antonio.maratea@uniparthenope.it

difficulties, RL is known and has been treated since the second half of the last century, with a rich literature in different domains (surveys in the context of data warehousing and for different phases can be found in *Brizan & Tansel (2006)* and *Christen (2011)*) and an established implementation process (see 'Phases of RL').

While data quality is related to and borrows techniques from RL and RD, it acts as an *a priori* filter, trying to prevent the insurgence of duplicate records with a proper control process *before* the data consolidation. RL and RD, on the contrary, act as *a posteriori* filters, trying to detect duplicate records and clean the data *after* consolidation. To the data analyst, they represent a required pre-processing step when it is legitimate to assume that the data to be analyzed are corrupted by consequence of a scarce quality control during acquisition, or come from heterogeneous sources with respect to time, place or context (please refer to *Batini & Scannapieco (2006)* for a general discussion of related issues).

The traditional approach to RL and RD is probabilistic, or rule-based, and only relatively recently Machine Learning alternatives have emerged (see *Zhao & Ram (2005)*). The probabilistic approach is grounded on the estimation of probabilities of match and on thresholds for the similarity scores; the rule-based approach tries to explicitly model the knowledge of domain experts; the Machine Learning approach, on the contrary, relies only on data and can be cost-sensitive, supervised, unsupervised, or semi-supervised.

In the following a multiple-criteria feature vector and a supervised classifier are proposed in the *classification* phase of the RL process, outperforming classic crude threshold-based methods and producing remarkable results.

The paper is organized as follows: in 'Background' the phases of a traditional RL process and the related literature on Neural Networks are sketched; in 'Materials and Methods' the proposed method is presented in detail; in 'Experiments' the used data and the experiments are fully described; finally, in 'Conclusions', main conclusions are drawn.

## BACKGROUND

First the currently established phases of a RL process will be outlined, then the recent literature on Neural Networks applied to RL will be summarized.

### Phases of RL

The RL of two sources generally involves five independent phases, each exploiting different techniques, criteria, and algorithms *Christen (2012)*:

1. *Data pre-processing*: the two sources are cleaned and normalized to ensure that all the data have the same format and are as much as possible error free;
2. *Indexing*: to save time, the record pairs that are evidently different are filtered out from the comparisons through clustering, sorting or blocking. Record Linkage is exponential in nature, as each record from the first source should be compared with all the records from the other sources, hence indexing is critical for performance;
3. *Comparison*: only the record pairs within the same block (or cluster) are actually compared one by one, using multiple similarity criteria that are conveyed into a similarity vector;

4. **Classification**: the similarity vector obtained from each pair of records within the same block (or cluster) is classified into one of three groups:

   - (M) –*matches*, that is pairs that do refer to the same entity of the real world;
   - (NM) –*non-matches*, that is pairs that do not refer to the same entity of the real world;
   - (PM) –*potential-matches* or ambiguous pairs, that is pairs that can't be classified with sufficient precision or reliability;

5. **Evaluation**: the classified pairs are reviewed for final labeling. Specifically, the PM class is subject to a *clerical review*, where domain experts decide for each ambiguous pair if it is actually a match or not.

## Related work

Being a possible consequence of both a poor data quality process than natural differences in the data evolving over time, RL has been found in countless domains and tackled using many techniques and approaches. The related literature can be broadly split into *census*, *administrative* or *health* related applications, with a majority of works exploiting probabilistic methods. Artificial Neural Networks (ANN from now on) as classifiers are less in number and relatively recent.

Some of the main issues of administrative data linkage are privacy and confidentiality requirements and to the absence of common identifiers (much like health data. Please refer to *Harron et al. (2017)* for a discussion and to *Vatsalan, Christen & Verykios (2013)* for a taxonomy). If from one side epidemiological studies linking census or administrative data to diseases are pretty common, from the other side the specific case of linking census with financial data at the individual level is rare, if not absent, in the RL literature. The reason is that medical data are mostly held by public agencies that have a public interest in sharing their data and supporting medical research, while banks are private companies that keep their data safe and secure on their servers, having little interest in sharing their analysis.

Recent literature on Machine Learning applied to RL includes *Aiken et al. (2019)*, that compares probabilistic, stochastic and machine learning approaches, showing that supervised methods outperform unsupervised ones; *Dong & Rekatsinas (2018)*, that surveys state-of-the-art data integration solutions based on Machine Learning with a discussion of open research challenges; *Kasai et al. (2019)*, that leverages Deep Learning in a combination of Transfer and Active Learning aiming to save labeled data up to an order of magnitude; *Di Cicco et al. (2019)*, that presents an attempt of explainable Deep Learning exploiting LIME, a popular tool for prediction explanations in classification; *Hou et al. (2019)*, that propose a paradigm called ''gradual machine learning'' where data are labeled automatically through iterative factor graph inference, starting with the easiest instances and going up to the hardest ones.

A survey of the state of the art is beyond the scope of this paper, so this section will focus on ANN classifiers applied to classification in RL, in line with the recent explosion of ANN related literature (please see *Maratea & Ferone, 2018*).

One of the first attempts is apparently due to *Zhao & Ram (2005)*, that proposed a set of ensemble learning methods combining multiple base classifiers, including a backpropagation ANN. Records are compared through various similarity measures and classifiers are merged through Bagging, Boosting, Stacking, Cascading or cross-validated committees. Using 25,000 records (20,000 non-matching examples and 5,000 matching examples) from an application service provider (ASP) for the airline industry, the authors tried to identify the same customer that reserved different flights with different airline companies, reaching over 99% of accuracy. Authors warn the reader on the limited generalization of the experiments due to the "somewhat balanced" data used.

More recently, *Wilson (2011)* showed on the base of genealogical databases that the results obtainable from the probabilistic RL are easily improvable through one of the various available Machine Learning or ANN techniques, and that even a simple single layer perceptron network tends to outclassify the probabilistic approaches, reaching 95% of precision and 97.2% of recall compared to 72.5% precision and 91% recall of the probabilistic methods.

A singular case is the work *Siegert et al. (2016)* in the linkage of epidemiological cancer registries data previously pseudo-randomized through hashing and encrypted for privacy reasons. Features are extracted from the obscured data and used as they were a new coding of the records, then the classification is performed on these coded data. Similarly to *Zhao & Ram (2005)*, multiple classifiers and ensembles are tested, with many aggregation functions. Approximately 35,000 match pairs and 38,000 of not matches for a total of 73,000 pairs of records were manually classified from the North Rhine-Westphalia cancer registry in Germany and used as training-set for the supervised learning classifiers. The proposed ANN is structured with a single hidden layer of 60 neurons and a sigmoidal activation function. Among the three classifiers used, the one based on ANN provided better results in both precision and recall terms, reaching a 95.2% precision and 94.1% recall. Even in this case, data are artificially balanced.

*Subitha & Punitha (2014)* propose the use of Elman's neural networks to pair the medical records collected by hand from different hospitals and departments, achieving an accuracy of 85% and a recall of 98% with respect to fuzzy decision trees (75% and 95%) and decision trees (79% and 96%). The comparing phase was performed using only the normalized Levenshtein distance as the similarity criteria. The Elman Neural Network (ENN) is a particular type of neural network in which a layer of neurons called "context units" connected with a fixed weight of 1.0 both to the input and to the output of the hidden layer is added. In the context units, a copy of the last output of the hidden layer is saved to be used for subsequent inputs. In this way the network can maintain a sort of state, allowing tasks such as the prediction of sequences that go beyond the power of a standard multilayer perceptron.

*Mudgal et al. (2018)* present a general framework for the application of Deep Neural Networks (DNN from now on) to RL, stressing connections with Natural Language Processing: three type of problems are highlighted: structured, textual and dirty RL. Their goal is to illustrate the benefits and limitations of DL when applied to RL. An empirical

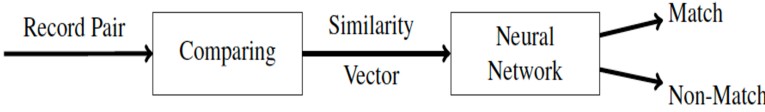

**Figure 1** Block diagram of the proposed method.

evaluation based on 4 models and 11 datasets is presented to show when DNN outperform traditional approaches, obtaining a relevant improvement in case of unstructured data.

*Kooli, Allesiardo & Pigneul (2018)* report a DNN application for RL using Relevant Word Extraction and Word Embedding instead of the classic calculation of the similarity vector from record pairs. Three architectures are compared: Multi-Layer Perceptron (MLP), Long Short Term Memory networks (LSTM) and Convolutional Neural Networks (CNN) on poorly structured scientific publications databases by getting very good results and showing improvements compared to classical similarity-based approaches. The authors point out that their approach is fully automatic unlike a previous job of *Gottapu, Dagli & Ali (2016)* that also uses DNN but in a human/machine hybrid fashion, by facilitating the manual categorization in a crowd-sourcing methodology, by proposing to the operator a list of possible matches. *Ebraheem et al. (2018)* used unidirectional and bidirectional recurrent neural networks (RNNs) with long short term memory (LSTM) hidden units to convert each tuple to a distributed representation, which was the base for subsequent classification. Experiments on 7 Datasets from different domains showed promising results.

# MATERIALS AND METHODS

A block representation of the proposed method is depicted in Fig. 1: it involves the *comparing* and *classifying* phases of RL (see 'Phases of RL'): it takes as input pairs of records and classifies them into "match" (M) or "non-match" (NM) through an ANN. The features of each candidate record pair are extracted by comparing each corresponding attribute with multiple similarity functions, resulting in a similarity vector (see below for more details). It is this similarity vector that is used for the subsequent classification of the candidate pair.

In order to have comparable tests, all preceding phases use the same methods and parameters.

## Data sources

Real data from separate municipal and banking databases have been gathered, chosen in the same geographical area to increase the probability of match (see Table 1). For both municipal and banking databases, each person is identified through the fiscal code (alias social security number, alias insurance number). Being a natural key and a reliable field, this common identifier was leveraged to create, through joins, a gold standard for training and evaluation purposes: if in a pair, both records had the same value for the fiscal code, they are considered a true-match.

The fiscal code is derivable from first name, last name, date and place of birth plus some special codes. Surprisingly, clerical reviews showed that in rare cases even the fiscal code

**Table 1 Used databases flanked by their size.**

| Database name | type | Alias | # record |
|---|---|---|---|
| ANAG_UNO | Municipal registry | **A** | 42,698 |
| ANAG_DUE | Municipal registry | **B** | 45,100 |
| ANAG_TRE | Municipal registry | **C** | 42,559 |
| ANAG_QUATTRO | Municipal registry | **D** | 57,434 |
| BCC_UNO | Bank details | **X** | 1,052,737 |
| BCC_DUE | Bank details | **Y** | 101,651 |
| BCC_TRE | Bank details | **Z** | 93,179 |

**Table 2 Attributes in common between banking and municipal databases.**

| Attribute | Municipal column | Bank column |
|---|---|---|
| SURNAME | COGNOME | INTESTAZIONE_A |
| NAME | NOME | INTESTAZIONE_B |
| SEX | SESSO | SESSO |
| BIRTH STATE | STATO_NASCITA | NAZIONALITA |
| BIRTH DATE | DAT_NASCITA | DATA_NASCITA |
| BIRTH PLACE | COM_NAS | DESCR_COM_NASC |
| BIRTH PROVINCE | PROV_COM_NAS | PROVINCIA_NASC |
| ADDRESS | VIA_DOM<br>NUMERO_CIVICO_DOM | VIA_RES |
| ZIP CODE | CAP_DOM | CAP_RES |
| PROVINCE | PROV_COM_DOM | PROV_RES |
| TELEPHONE | TELEFONO | NUMERO_TELEFONO |

presents some errors, i.e., does not correspond to the value derivable from the other fields of the record. This can lead to rare cases in which a pair of records is correctly classified as a match by the classifier but results indeed a false positive for the evaluation metric. For this reason, all the figures of merit presented hereafter have to be considered conservatively estimated.

## Relevant attributes

As a first step, it is necessary to select a subset of relevant and shared attributes between the municipal and banking databases to be used in the RL process. The selected attributes are shown in Table 2.

Special attention should be paid to attributes *address*, *zip code* and *province* because they have a different meaning, representing the home address in municipalities databases and the residence address in banking databases. These values are often the same but do not always coincide.

Each attribute is cleaned, standardized and normalized through multiple transformations, turning everything lower case, removing special symbols, punctuation, repeated spaces, non-alphabetic characters and finally normalizing accented letters.

### Indexing

Indexing is performed with *multiple blocking indexing*, combining the results obtained from 3 blocking keys.

1. Blocking key 1: <SURNAME, NAME, B.DATE>, using the Double Metaphone phonetic algorithm for the key comparison;
2. Blocking key 2: <B.DATE, B.PLACE, B.PROVINCE>, using the Double Soundex phonetic algorithm for the key comparison;
3. Blocking key 3: <SURNAME, NAME, SEX, B.DATE, B.PLACE, B.PROVINCE> using for the key comparison: the last 3 character suffix for name and surname; the first 4 character prefix for the birth place and province; and the year and month for the birth date.

These blocking criteria were determined through experimental test and chosen to maximize the pair completeness as much as possible by keeping the number of candidate record pairs generated low, to reduce the execution time in view of the numerous tests to be performed.

### Comparing

In the comparison phase, the similarity of each pair of records $(A, B)$ is measured using a set $\mathcal{S}$ of string-based similarity functions on the corresponding attributes (name of the first record with name of the second record, surname of the first record with surname of the second record and so on). Each comparison function has a normalized output in the interval $[0, 1]$. Where 0 indicates maximum dissimilarity and 1 indicates maximum similarity (please see *Navarro, 2001*; *Christen, 2012*).

The set $\mathcal{S}$ of comparison functions used is listed below:

(a) Jaro–Winkler, (b) Levenshtein, (c) Cosine, (d) Jaccard, (e) Siresen-Dice, (f) Bigrams, (g) Trigrams, (h) Exact.

Each one of the corresponding attributes pairs $(a_i, b_i)$ is compared using all the function in the set and the resulting values are then chained in a similarity vector **s**.

$$\mathbf{s} = \text{sim}_{\mathcal{S}}(a_1, b_1) \oplus \text{sim}_{\mathcal{S}}(a_2, b_2) \oplus \text{sim}_{\mathcal{S}}(a_3, b_3) \oplus \dots$$

Figure 2 shows an example of this procedure. At the end, each pair of records will be associated with a similarity vector to be used for classification.

### Classifying

The ANN used for the classification is shown in the Fig. 3, it's a fully connected MultiLayer Perceptron network (MLP from now on), with two hidden layers in a pyramidal structure, a *ReLu* activation function:

$$\text{ReLu}(x) = x^+ = \max(0, x)$$

and a *Softmax cross entropy* loss function:

$$\mathcal{L}(y, \hat{y}) = \sum_i H(y_i, \hat{y}_i) = \sum_i y_i \cdot \log \hat{y}_i.$$

The optimal number of neurons and the size of the layers have been determined through iterative optimization on experimental tests. The final architecture is:

$$similarity(A, B) = (0.833, 0.571, 0.288, 0.961, 0.666, 0.777, 0.774, 0.962,$$
$$1.000, 1.000, 1.000, 1.000, 0.333, 0.333, 0.000, 0.333,$$
$$1.000, 1.000, 1.000, 1.000, ..., 0.000, 0.000, 0.000, 0.000)$$

**Figure 2** Example of the similarity vector obtained comparing two records using four similarity functions (please see Table 2 for attribute mapping).

- *Input Layer*: a layer of as many neurons as the components of the similarity vectors to be classified;
- *Hidden Layer*: two layers to form a pyramidal structure, the first with 8 and the second with 4 neurons;
- *Output Layer*: a layer of two neurons, one for the class of the matches (M) and the other for the non-matches (NM).

For the initialization the `glorot_uniform_initializer` (aka Xavier uniform initializer) was used. With this random initalization of the MLP parameters no relevant changes in the performance were noted over different runs.

The network was trained using the Adam optimizer by applying a L1 regularization to avoid overfitting.

### Training data-set generation

The training data-set, in the format (*feature*, *label*) is generated based on the candidate record pairs identified in the indexing phase between databases *A* and *B*, where:

- *feature*: is the similarity vector obtained comparing the records of the pair;
- *label*: is true-match (M) or true-not match (NM), according to the gold standard.

The built training data-set contains 10,876 samples, 1,567 of which are labeled as true-match (M) and 9300 as true-non-match (NM).

Since non-matching record pairs are more than matching ones, the training data are moderately imbalanced (they are in the same order of magnitude). Over-sampling techniques like ADASYN (*He et al., 2008*) and based on SMOTE, such as SMOTENC (*Bowyer et al., 2011*) SMOTENN (*Batista, Prati & Monard, 2004*), SMOTETomek (*Batista,*

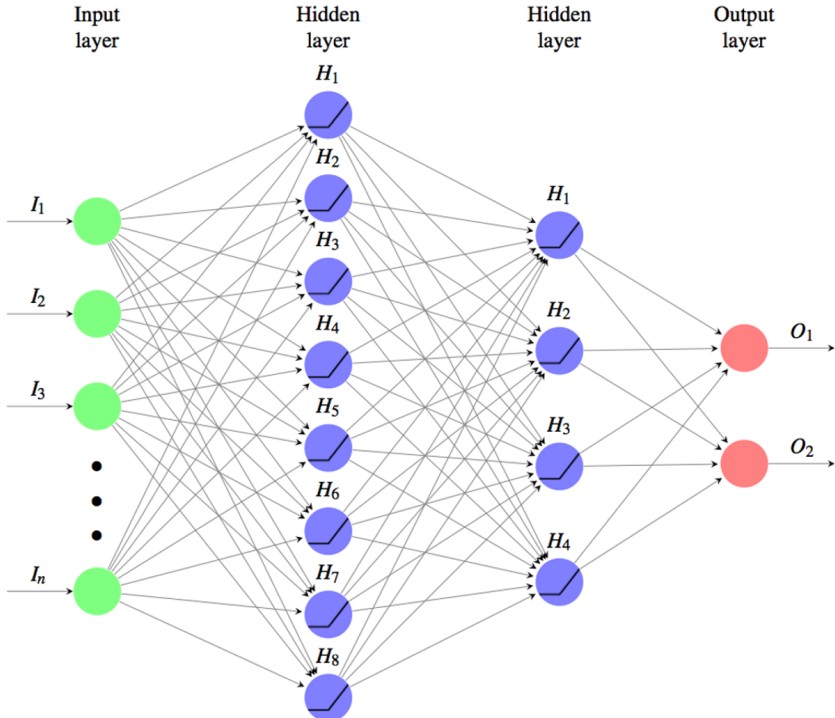

**Figure 3** **Used fully connected MLP architetture for the classifyng phase.** Where the input layer as many neurons as the components of the similarity vectors to be classified. There are two hidden layers one of size 8 and the other of 4 and the output layer is composed of two units, one for the class of the matches (M) and the other for the non-matches (NM). The activation function used is the ReLu and the loss function is the Softmax cross entropy. The weights are initalized with Xavier uniform initializer and the training was performed using the Adam optimizer by applying a L1 regularization.

*Bazzan & Monard, 2003*) have been tested, without any significant improvement, so oversampling was skipped from final experiments.

## EXPERIMENTS

For each test the same set of attributes, pre-processing and indexing techniques have been used in order to focus on the comparing and classifying phases.

Figure 4 shows the starting condition for the tests in order to have a reference on the number of true-match and of pairs identified by the indexing between the various coupled databases.

Since there are seven different databases available, grouped in municipalities (four) and banks (three), 21 executions will be performed for each test, pairing databases between groups and excluding deduplication (that is the match of a database against itself). Only the pair (A, B) is used as training-set.

The chosen figure of merits are *precision* and *recall*, preferred to plain accuracy due to the moderate imbalance, as explained by *Christen & Goiser (2007)*. Their average and standard deviation over the 21 runs are reported in the following.

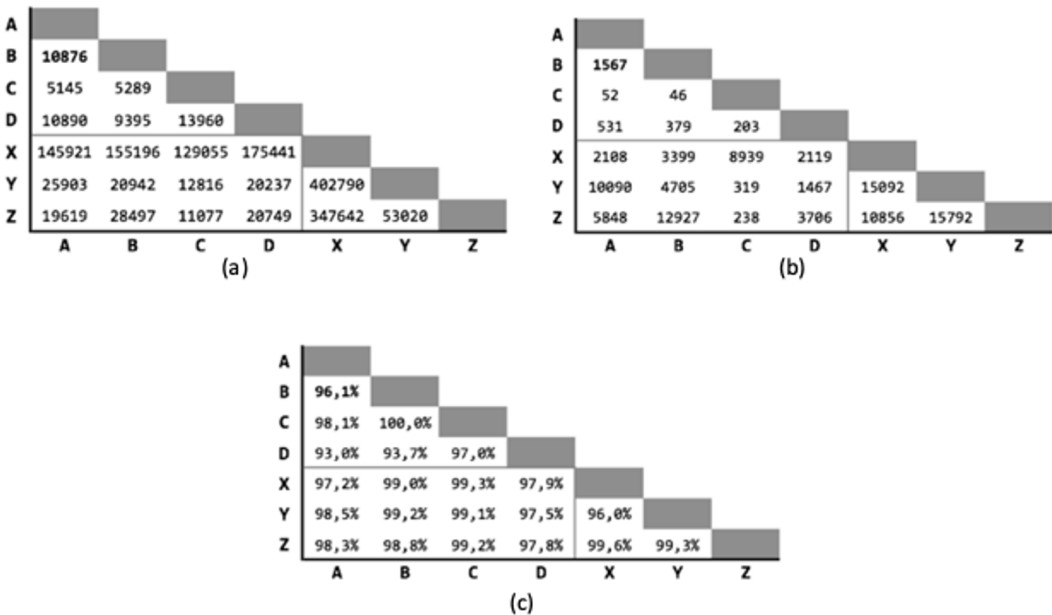

**Figure 4** **Tests starting conditions after the indexing phase, for all the possible pairs of databases.** (A) Number of candidate record pairs to be classified, obtained from the indexing phase. (B) Number of True-Match record pair in the gold standard. (C) Pair completeness, i.e., percentage of true match retrieved in the indexing phase with respect to the gold standard.

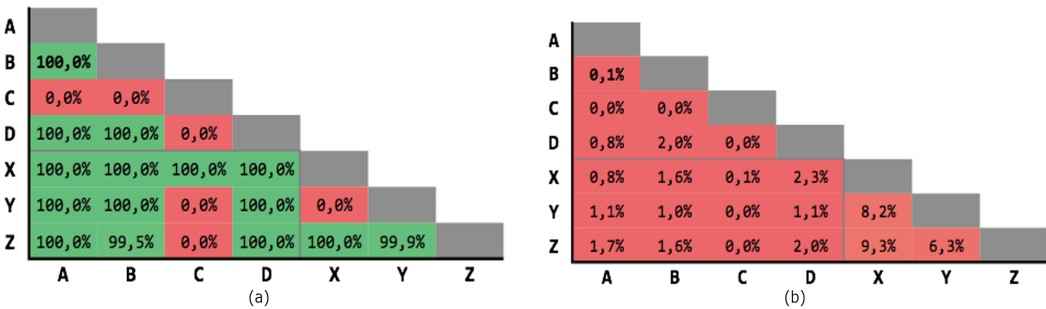

**Figure 5** **Exact matching results.** (A) Precision; (B) recall.

## Exact matching

In the very first test, RL has been performed with an *Exact Match* goal, where the candidate record pairs are classified as match only if all the respective fields are perfectly equal. This test has been carried out only to show how much two databases, while containing the same information, actually differ in the values of their attributes.

As expected (Fig. 5), the precision is extremely high, but the number of matches is extremely low, as shown by the recall. In fact, in some cases no pair have been identified, with a maximum of only 4 matches, by consequence of errors, noise, and random variations in the corresponding data.

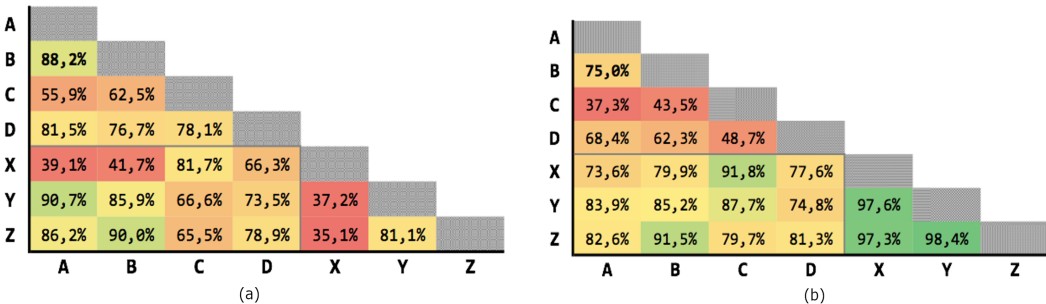

**Figure 6** Results for Levenshtein normalized distance with threshold $\theta = 0.7$. (A) Precision; (B) recall.

## Threshold based classification

In the next test an *optimal binary threshold classifier* was used. The threshold value was determined through a PR graph (precision–recall) generated on the training data-set. As threshold, the value having coordinates closest to the ideal point (1.0,1.0) was chosen (see Figs. A1–A3).

### Levenshtein threshold

in this test, the only similarity criterion was the Levenshtein normalized distance (*Levenshtein, 1966*)—one of the most widely used comparison metrics.

- *Comparing*: Levenshtein normalized distance.
- *Classifying*: binary threshold with $\theta = 0.7$ as optimal value, applied to the weighted sum of the similarity vector.
- *Weighting*: each attribute has the same importance and weight.

Figure 6 shows the results obtained on the 21 executions. The average precision is 69,63% and the average recall is 77,04%, although with high variability.

### Multiple criteria threshold

In this test, multiple comparison criteria for each attribute were used, with the underlying idea that different criteria measure different facets of the similarity between them. Recall actually improved.

- *Comparing*: each attribute is compared using the following metrics: (a) Jaro–Winkler, (b) Levenshtein, (c) Cosine, (d) Jaccard, (e) Siresen-Dice, (f) Bigrams, (g) Trigrams, (h) Exact.
- *Classifying*: binary threshold with $\theta = 0.63$ as optimal value applied to the weighted sum of the similarity vector.
- *Weighting*: each attribute has the same importance and weight.

Figure 7 shows the results of the 21 executions: the average precision is 71,93% and the average recall is 85,85%, although with high variability.

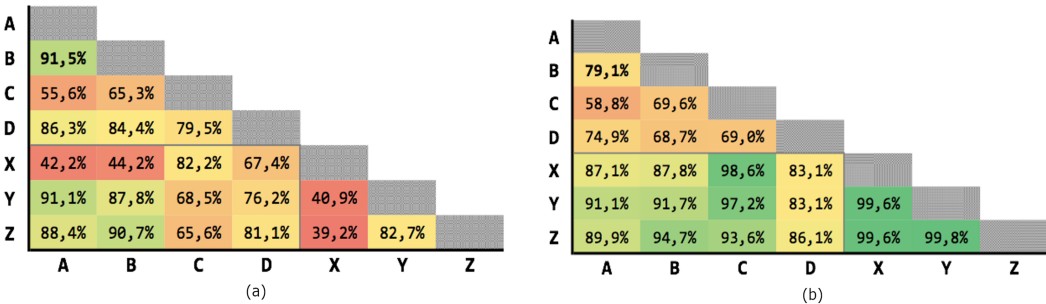

**Figure 7   Results for multiple criteria similarity with threshold $\theta = 0.63$, unweighted.** (A) Precision; (B) recall.

### Weighted multiple criteria threshold

In this test, the previous results were improved by varying the importance of the various fields through an appropriate weighing. The weights, for each attribute, were automatically determined based on their distinct values distribution, and normalized in such a way that their sum is equal to 1.0 (see Fig. A4).

Among the various possibilities of normalization (*linear*, *max*, *quadratic*, *exp* ...) that *logarithmic* seems to give the best results for both overall precision and recall maintaining low variance (see Fig. A4).

- *Comparing*: each attribute is compared using the following metrics:
  (a) Jaro–Winkler,
  (b) Levenshtein,
  (c) Cosine,
  (d) Jaccard,
  (e) Siresen-Dice,
  (f) Bigrams,
  (g) Trigrams,
  (h) Exact.
- *Classifying*: binary threshold with $\theta = 0.63$ as optimal value applied to the weighted sum of the similarity vector.
- *Weighting*: the weights of the various attributes are estimated according to the distribution of their distinct values and normalized using a logarithmic function. The associated weight is directly proportional to the number of distinct values over the totals.

Figure 8 shows the results of the 21 executions: the average precision is 89,37% and the average recall is 94,74%, with low variability.

## MLP based classification

To allow a fair comparison, the tests using MLP classifier have followed the same schema of the the previous ones.

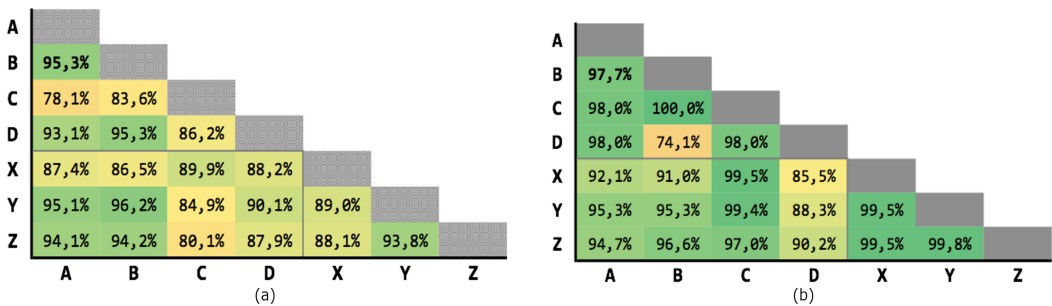

**Figure 8** **Results for multiple criteria similarity with threshold $\theta = 0.63$, weighted.** (A) Precision; (B) recall.

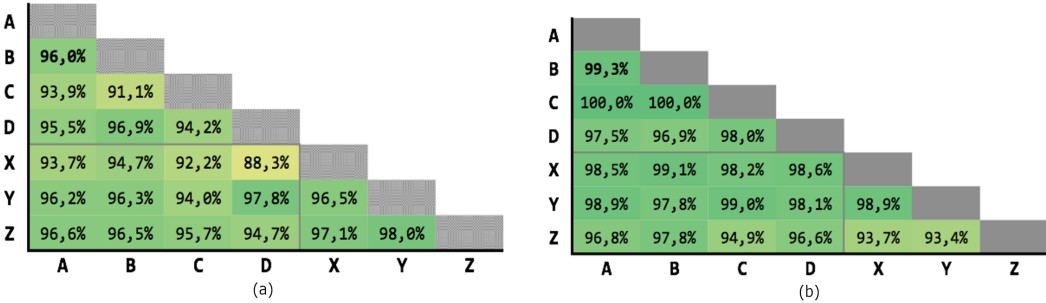

**Figure 9** **Results of the Levenshtein test with MLP classifier.** (A) Precision; (B) recall.

### MLP Levenshtein

In this test, only the normalized Levenshtein distance was used, likewise the homonym test with the threshold classifier. As can be seen, the results clearly outperform all previous ones.

- *Comparing*: Levenshtein normalized distance only.
- *Classifying*: MLP based classifier.

Figure 9 shows the results of the 21 executions: the average precision is 95,04% and the average recall is 97,71%, with very low variability.

### MLP with multiple criteria

In this test, multiple comparison criteria for each attributes have been used, likewise the homonym test with the threshold classifier. The results are almost perfect, especially for recall. In addition, the high precision allowed the manual control of the "false-positive" pairs, many of which are actually correct, but due to errors have a different fiscal code in the gold standard (see *Data sources*). Considering these fixes, precision reaches 100% in some cases.

- *Comparing*: Each attribute is compared using the following comparators:
  (a) Jaro–Winkler,

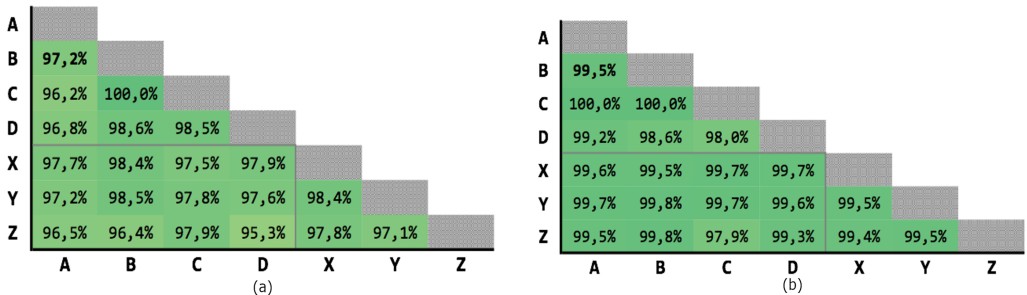

**Figure 10  Results of test #2 for MLP classification.** (A) Precision; (B) recall.

**Table 3  Experimental results Summary: summarized results obtained through the mean values and standard deviation of the 21 executions for each test.**

| Classifier | Comparators | %Precision | %Recall |
|---|---|---|---|
| Threshold | Levenshtein | $69{,}63\% \pm 17{,}76\%$ | $70{,}04\% \pm 16{,}66\%$ |
| Threshold | All | $71{,}93\% \pm 17{,}51\%$ | $85{,}85\% \pm 11{,}61\%$ |
| Weighted Threshold | All | $89{,}37\% \pm\ 4{,}98\%$ | $94{,}74\% \pm\ 6{,}12\%$ |
| Multilayer Perceptron | Levenshtein | $95{,}04\% \pm\ 2{,}28\%$ | $97{,}71\% \pm\ 1{,}78\%$ |
| Multilayer Perceptron | All | $97{,}58\% \pm\ 0{,}99\%$ | $99{,}39\% \pm\ 0{,}55\%$ |

  (b)  Levenshtein,

  (c)  Cosine,

  (d)  Jaccard,

  (e)  Siresen-Dice,

  (f)  Bigrams,

  (g)  Trigrams,

  (h)  Exact.

- *Classifying*: MLP based classifier.

Figure 10 shows the results of the 21 executions: the average precision is 97,58% and the average recall is 99,39%, with very low variability.

## Summary results

In Table 3, a comprehensive view of the obtained results through the mean values and standard deviation of the 21 executions is reported.

## CONCLUSIONS

First the various stages of the classic Record Linkage (RL) process have been presented, then a classifier based on multiple criteria and Neural Networks has been proposed in the *classification* stage of RL. Specifically, the chaining of different similarity measures on different fields has been used as feature vector for the subsequent classification of record pairs based on Multi-Layer Perceptron (MLP). The proposed feature vector plus MLP classifier has been tested on several real-world datasets belonging to geographically close

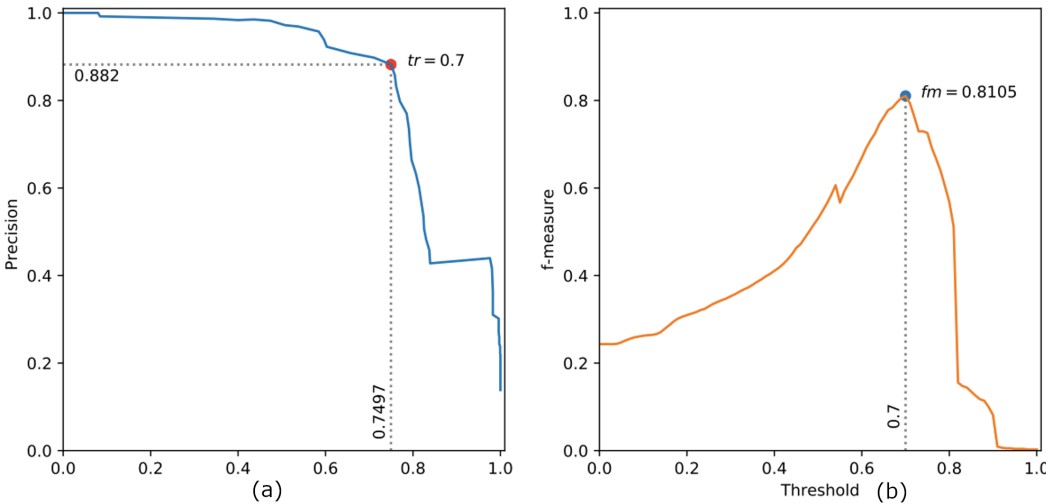

**Figure A1  Levenshtein threshold.** (A) The precision–recall curve generated on the training-set in experiment #1; The optimal threshold is $tr = 0.7$. (B) The F1-score when the threshold changes.

banks and municipalities, scoring remarkably (please see Table 3) and clearly outperforming the threshold-based methods. Neural Networks, even with a shallow architecture and few nodes, proved to be effective classifiers and should be seriously considered for RL when even a modest amount of labeled data is available.

## ACKNOWLEDGEMENTS

The authors thank SADAS s.r.l. (https://www.sadasdb.com/en/) for providing the necessary data and making their columnar database, proprietary technologies and research laboratories available.

## APPENDIX

### Precision recall curve

In this section are shown in Figs. A1–A3, for each threshold-based test, the precision–recall curve generated on the training-set database pair (A, B). These curves were used to determine the optimal threshold value for classifiers.

### Weight vector estimation

In this section, the methodology used in experiment 3 for the weights estimation is described. The weight vector has as many components as there are attributes selected for the RL each of which is calculated as:

$$w_i = \frac{\log\|\mathrm{supp}\,\mathbf{a}_i\|}{\log\|\mathbf{a}_i\|} \cdot \frac{\log\|\mathrm{supp}\,\mathbf{b}_i\|}{\log\|\mathbf{b}_i\|} \tag{1}$$

where $\mathbf{a}_i$ and $\mathbf{b}_i$ are the multi-set values of the corresponding i-th attribute of the first and second table respectively. The notation supp· indicate the support, i.e., the set of unique items in multi-set and $\|\cdot\|$ denotes the cardinality.

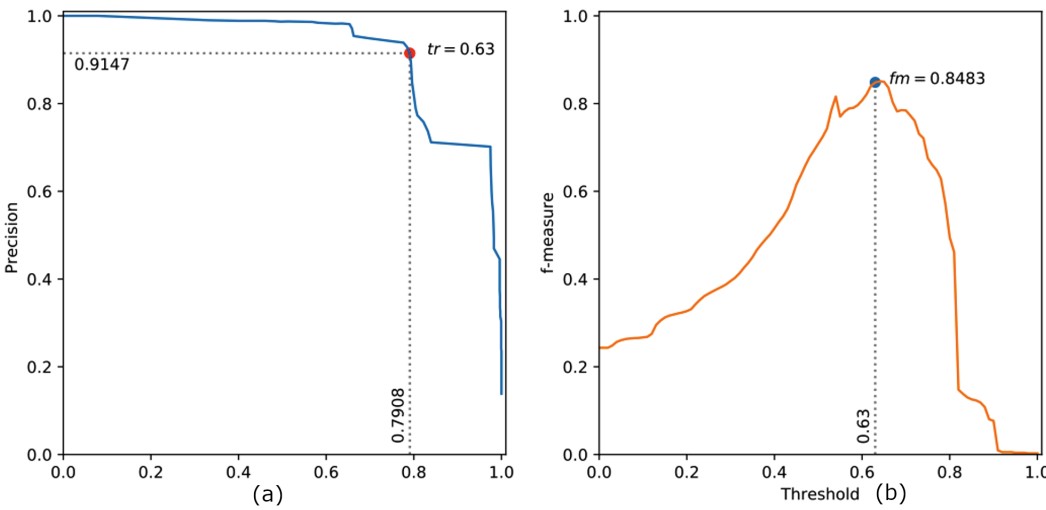

**Figure A2** **Multiple criteria threshold.** (A) The precision–recall curve generated on the training-set in experiment #2; the optimal threshold is $tr = 0.63$. (B) The F1-score when the threshold changes.

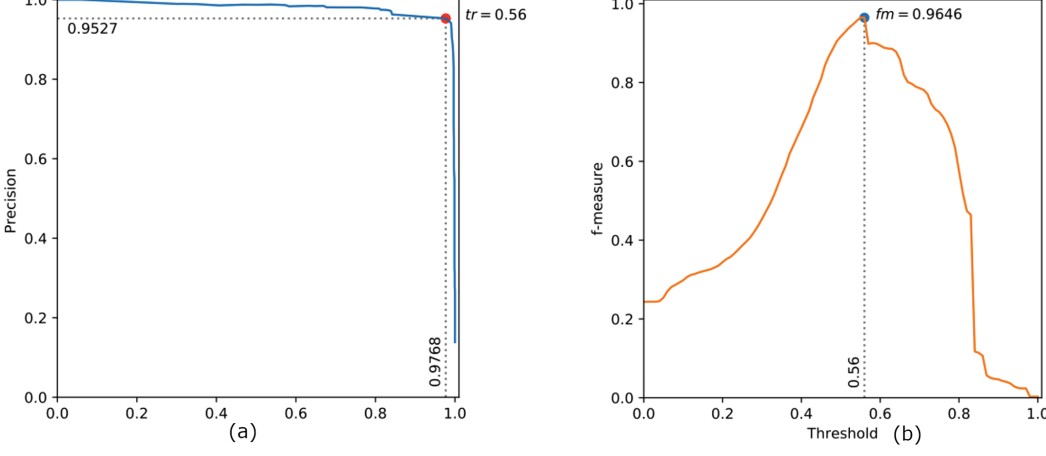

**Figure A3** **Weighted multiple criteria threshold.** (A) The precision–recall curve generated on the training-set in experiment #3; the optimal threshold is $tr = 0.56$. (B) The F1-score when the threshold changes.

This measure takes into account both the number of distinct values and the size difference of the two tables.

## Weight vector normalization

In this section are shown, the vector normalization technique and then, for each normalization base function, in Fig. A4, the precision–recall curve generated on the training-set database pair (A,B). These curves were used to select the best type of normalization.

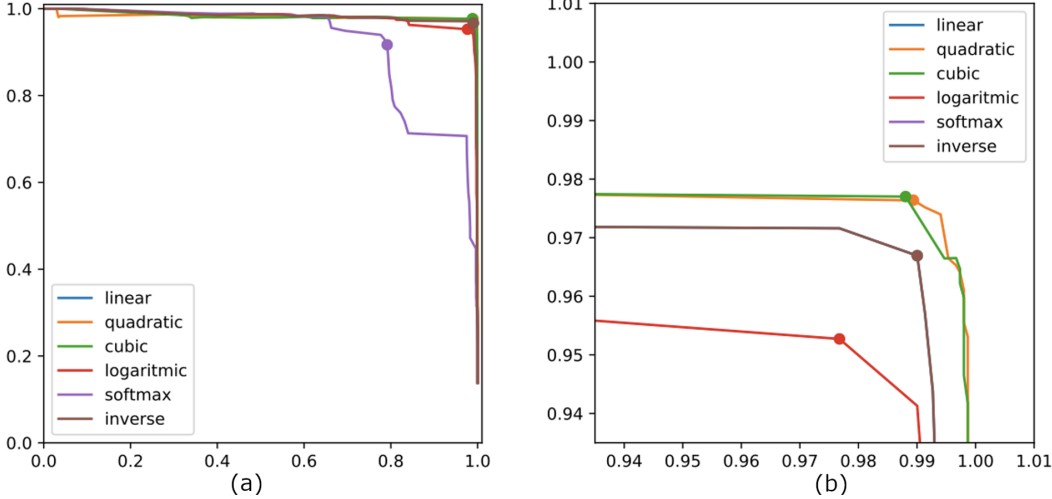

**Figure A4** (A) **The precision–recall curves generated on the training-set in experiment #3 for each of the normalization techniques; (B) a zoomed view of the same curves.** The relative winner, in terms of precision and recall, was the quadratic normalization, but results of higher quality and lower variance were obtained using a logarithmic function.

### Normalization

To alter the contribution of the various attributes during the classification while maintaining unchanged the weighted sum of the components of the similarity vector, a normalization of the weight vector is necessary.

Given a weight vector $\mathbf{w} = (w_1, w_2, \ldots, w_n)$ and same *base* function $f$ a normalized vector $\hat{\mathbf{w}} = (\hat{w}_1, \hat{w}_2, \ldots, \hat{w}_n)$ can be obtained simply by applying element-wise the equation:

$$\hat{w}_i = \frac{f(w_i)}{\sum_{j=1}^{n} f(w_j)} \tag{2}$$

i.e., applying the function $f$ to each element of the input vector $w_i$ and normalizing these values by dividing by the sum of all these values; this normalization ensures that the sum of the components of the output vector $\hat{\mathbf{w}}$ is 1.

The tested base function $f$ are listed below:

| | | | |
|---|---|---|---|
| linear: | $f(x) = x$ | quadratic: | $f(x) = x^2$ |
| cubic: | $f(x) = x^3$ | logaritmic: | $f(x) = \log(1+x)$ |
| softmax: | $f(x) = e^x$ | inverse: | $f(x) = x^{-1}$ |

### Precision–recall curves

Results for both overall precision and recall among the various possibilities of normalization: *linear, max, quadratic, exp ... logarithmic.*

### Funding
The authors received no funding for this work.

### Competing Interests
The authors declare there are no competing interests.

### Author Contributions
- Antonio Maratea conceived and designed the experiments, analyzed the data, authored or reviewed drafts of the paper, and approved the final draft.
- Angelo Ciaramella analyzed the data, authored or reviewed drafts of the paper, and approved the final draft.
- Giuseppe Pio Cianci performed the experiments, authored or reviewed drafts of the paper, performed the computation work, prepared figures and/or tables, and approved the final draft.

### Data Availability
 The data, code, and a readme file are available in the Supplementary Files.

### Supplemental Information
Supplemental information for this article can be found online at http://dx.doi.org/10.7717/peerj-cs.258#supplemental-information.

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
