# Peer review of "Record linkage of banks and municipalities through multiple criteria and neural networks"

_PeerJ Computer Science, doi:10.7717/peerj-cs.258_

## Round 0.1 · original submission · Major Revisions

Three review reports have been received. The reviewers have raised many good points. Please give a point-to-point response.

·

Basic reporting

This paper analyzes several strategies for linking records in different dataset. The paper is well presented, the experimental methodology is appropriate, and the obtained results are interesting.

Experimental design

The experimental design is appropiate and the obtained results are interesting and significant.

Validity of the findings

From my point of view the main contributions are:
- Good state of the art
- Interesting analysis of several strategies comparing between them
- Good results with the architecture proposed.

Additional comments

This paper analyzes several strategies for linking records in different dataset. The paper is well presented, the experimental methodology is appropriate, and the obtained results are interesting.
My comments to improve the paper are:
- The background section is very interesting and complete. Regarding the related work, my suggestion would be to include more references from the last 2-3 years. Some of them are very all.
- About the data description, I’d like to ask if this data is available and where?
- Line 167: All figure of merits -> All figures of merit??
- When describing the MLP, what initialization have you used?
- For the experiments you have used precision and recall. I’d suggest using (f_measure as in the annex) that combines both and the comparison is easier.
- I’d like more explanation about table 3. What is column #? How did you compute the confidence intervals? You said these number are variability but how is it computed?

·

Basic reporting

Generally good English, with good citations to relevant sources.

Here are minor updates that should be made or considered:

26, 28 – The word “quest” seems awkward. Consider using “process of identifying” instead.

35 – “multi-nationality grow” => “number of nationalities grows”

35 – Consider replacing the semicolon (;) with a period, and starting a new sentence.

46 – Instead of “inhomogeneous”, you could use “heterogeneous”

52 – add commas: “approach, on the contrary, relies…”

55 – By “threshold-based methods”, do you mean “probabilistic methods?” Because a neural network still uses thresholds on its final classification output—it just uses trained weights instead of observed statistics to come up with the value.

85 – I don’t agree that record linkage is solely needed because of “a poor data quality process.” In many cases, different sources gathered different data according to their own needs, and may have done so with very high quality; and yet someone else later decided to combine data from these sources together and try to decide which entities are the same. A health care system will gather more health data; and a banking system will gather more financial data; and a travel system may include passport data. Even if these are all done with high quality, the failure to have the same fields in each system is not necessarily a “quality” problem as much as a natural difference in original intent. Similarly, a female changing her name when she gets married doesn’t constitute a quality problem, but, rather, a natural name variation that occurs over time. I do agree that quality is one factor, but natural differences in the original intent of the data capture is another.

90 “Some of the main issues of administrative data linkage are privacy…” (Add “Some of the” and get rid of “due to”, as that is awkward).

95 – “hold” => “held”

101 – “among which” => “including”

111 – The precision and recall numbers mentioned there are somewhat meaningless without listing the P/R of the probabilistic methods they are being compared to. Finish the sentence with “…compared to x% precision and y% recall of the probabilistic methods.”

139 – “Memorynetworks” => “Memory networks”
141 – classics => classical
143 – uses also => also uses
145 – use => used
146 – which is => which was
147 – show => showed
152 – “each corresponding attributes” => “each corresponding attribute”

191: Figure 2 is never referenced in the text.

200: “as loss function” => “loss function” (don’t need “as” because you have “a” earlier)

202: “the experimental tests” => “experimental tests”

217: Seems like you need spaces between SMOTE and the authors’ names. (“SMOTENCBowyer et al (2011)” => “SMOTE-NC (Bowyer et al., 2011)”)

229: I don’t understand. Do you mean “due to the moderate imbalance, as explained by Christen and Goiser (2007)”?

244: Need a space between your text and the reference (“distanceLevenshtein”)

Experimental design

The experiments seemed reasonably well designed. The following are simply suggestions that the authors may want to consider either commenting on in the paper, or perhaps updating their experiments (for this paper or in future work).

153: In addition to comparing corresponding attributes with multiple similarity functions, have you considered higher-level combinations of attributes? This may not always make sense, but with some data sources, you can have higher-level features beyond direct attribute comparisons, like when one person has a birth place available in the data and the other doesn’t; but the other person has a residence place but the first doesn’t. Then a feature that says “Do any of the places overlap?” would not use corresponding attributes, but can still be helpful. Just a thought to consider.

158: Is the final goal of the classifier to be used only on pairs within the same geographical area? If so, then this is fine. If not, then there may well be a “hole” in the classifier: People can move from one area to another, and if the classifier is meant to discover those pairs as well, then the training data may fail to cause these cases to work well, and the evaluation data may fail to alert you to this gap.

168: It would be good to look at all cases in the gold standard data where the fiscal code mismatched but the classifier says it’s a match and correct any that are auto-labeled incorrectly. (And vice-versa: Cases where the fiscal code says it’s a match but the classifier disagrees strongly). Correcting your training data takes time, but is usually needed for world-class accuracy.

193: Using string-based similarity metrics on dates is likely not the best thing to do. You should really parse the dates to get year, month and day; and then compare those values. Being a couple ‘characters’ off on the day is less severe than being off on the year, for example. You might consider looking at your data to analyze how often matches and non-matches disagree on dates, and by how much. Are matches sometimes off by a few days? Or are typos the largest source of differences? You can sometimes use a higher-level feature like “Date is within 1 day; within 1 week; within 1 year; greater than 1 year difference” if date differences are possible on matching records, and have each of these features be a neural network weight, rather than having just one feature per attribute.

Similarly, place standardization should tell you that “IT” and “ITALY” are the same place, whereas “AL” and “ITALY” are not, even though both have only a two character overlap.

Validity of the findings

The conclusions were not surprising, but were well stated. I was a little confused at first by the table of counts and results, but I think I eventually realized that the counts and accuracies for each cell in the tables in Figure 4 and beyond were those pairs that were found in the original blocking stage. You might want to state that.

Additional comments

Overall, this seemed like a fairly straightforward application of neural networks to a record linkage problem using traditional multi-pass blocking and simple field comparison features with a list of string comparison functions for field comparison. I would guess that doing actual date and place standardization would improve results, rather than doing generic string-matching on these kinds of fields.

Reviewer 3 ·

Basic reporting

Clear, unambiguous, professional English language used throughout.

Experimental design

In lines 23-24, you say your method scored remarkable results compared to threshold-based method
Qs: How does your method perform against the other methods, i.e. the rule-based approach?

In lines 157-158, You say, " databases have been gathered, chosen in the same
geographical area to increase the probability "

Qs: Please can you elaborate more on the challenges arise when using databases from different geographical areas?

In lines 188- 189 you say "These blocking criteria were determined through experimental test and chosen to maximize the pair completeness as much as possible by keeping the number of candidate record pairs generated low, to reduce the execution time in view of the numerous tests to be performed."

Qs: Can you please give more details about how you conducted this experimental test?

Validity of the findings

If there is a weakness in the manuscript, it's because the findings are based on using data from the same geographical area and relatively small compared to other studies mentioned in lines 100 - 121.

Additional comments

1) Figure3 and the description of the ANN in lines 200 - 207 is too generic. I suggest you provide more details on the input size, structure of the hidden layers, etc.

---

## Round 0.2 · accepted · Accept

I am glad to recommend publication based on the reviewers' assessment.

·

Basic reporting

The authors have addressed propoerly my comments so I think the paper is ready for publication.

Experimental design

The authors have addressed propoerly my comments so I think the paper is ready for publication.

Validity of the findings

The authors have addressed propoerly my comments so I think the paper is ready for publication.

Additional comments

The authors have addressed propoerly my comments so I think the paper is ready for publication.

·

Basic reporting

The updated paper fixed all of the minor English errors I observed. The paper is well written with clear references to figures and tables.

Experimental design

The paper presents a somewhat straightforward application of machine learning to a record linkage problem. The experimental design is solid, and the paper can serve as a decent example of approaching such a problem.

Validity of the findings

The findings seem valid. The explanation by the author of the choices they made in terms of features, etc., satisfy my concerns.

Reviewer 3 ·

Basic reporting

Clear and unambiguous, professional English used throughout.
Literature references, sufficient field background/context provided.

Experimental design

Original primary research within Aims and Scope of the journal.

Validity of the findings

Conclusions are well stated, linked to original research question & limited to supporting results.

Additional comments

I am satisfied with the provided answers. Please incorporate them within the manuscript.